# Comparing Long-Term Prognosis in Chronic Critically Ill Patients: A Case Series Study of Medical versus Surgical Sepsis

**DOI:** 10.3390/medicina59091617

**Published:** 2023-09-07

**Authors:** Benjamin Mancini, Jiabin Liu, Abigail Samuelsen, Judie A. Howrylak, Lisa Schultz, Anthony S. Bonavia

**Affiliations:** 1Penn State College of Medicine, 500 University Dr., Hershey, PA 17033, USA; 2Department of Anesthesiology, Critical Care & Pain Management, Hospital for Special Surgery, New York, NY 10021, USA; 3Department of Anesthesiology, Weill Cornell Medical College, New York, NY 10021, USA; 4Department of Anesthesiology and Perioperative Medicine, Division of Critical Care Medicine, Penn State Hershey Medical Center, 500 University Dr., Hershey, PA 17033, USA; 5Department of Medicine, Division of Pulmonary, Allergy and Critical Care Medicine, Penn State Hershey Medical Center, 500 University Dr., Hershey, PA 17033, USA

**Keywords:** sepsis, chronic critical illness, surgery, long-term outcomes

## Abstract

*Background and Objectives:* Chronic critical illness (CCI) is a syndrome characterized by persistent organ dysfunction that requires critical care therapy for ≥14 days. Sepsis and respiratory failure constitute the two primary causes of CCI. A better understanding of this patient population and their clinical course may help to risk-stratify them early during hospitalization. Our objective was to identify whether the source of sepsis (medical versus surgical) affected clinical trajectory and prognosis in patients developing CCI. *Materials and Methods:* We describe a cohort of patients having acute respiratory failure and sepsis and requiring critical care therapy in the medical (MICU) or surgical (SICU) critical care units for ≥14 days. Given the relative infrequency of CCI, we use a case series design to examine mortality, functional status, and place of residence (home versus non-home) at one year following their index hospitalization. *Results:* In medical patients developing CCI (*n* = 31), the severity of initial organ dysfunction, by SOFA score, was significantly associated with the development of CCI (*p* = 0.002). Surgical patients with CCI (*n* = 7) experienced significantly more ventilator-free days within the first 30 days following sepsis onset (*p* = 0.004), as well as less organ dysfunction at day 14 post-sepsis (*p* < 0.0001). However, one-year mortality, one-year functional status, and residency at home were not statistically different between cohorts. Moreover, 57% of surgical patients and 26% of medical patients who developed CCI were living at home for one year following their index hospitalization (*p* = 0.11). *Conclusions:* While surgical patients who develop sepsis-related CCI experience more favorable 30-day outcomes as compared with medical patients, long-term outcomes do not differ significantly between groups. This suggests that reversing established organ dysfunction and functional disability, regardless of etiology, is more challenging compared to preventing these complications at an earlier stage.

## 1. Introduction

Advances in the treatment of acute sepsis, such as early antibiotic therapy, infectious source control, and goal-directed fluid resuscitation, have reduced short-term mortality rates [1]. Nonetheless, around fifty percent of individuals with sepsis continue to undergo an extended period of illness marked by enduring organ dysfunction, frequent medical complications, and a survival rate of merely 63% at the end of six months [2]. Chronic critical illness (CCI) refers to the ongoing, life-threatening organ dysfunction that requires critical care for 14 days or longer [2]. The presence of sustained pro-inflammatory cytokine levels and a lack of anti-inflammatory processes to modulate and repair tissues are key features of this condition [3,4]. Given the medical complexity of these patients and the relative infrequency of this syndrome, it is challenging to gain a comprehensive understanding of their clinical trajectories and clinical outcomes.

In 2009, the healthcare costs associated with CCI amounted to USD 26 billion, and over 60% of these patients had sepsis upon ICU admission [5]. In fact, sepsis is the second leading cause of CCI in the United States, with prolonged acute mechanical ventilation being the first [5]. CCI patients comprise only 5% of ICU admissions, yet they account for over 30% of ICU bed days and more than 14% of hospital bed days [6]. Despite the significant financial burden of caring for patients with CCI, long-term outcomes data are sparse but generally understood to be poor [5,7,8]. The clinical and financial burden of CCI patients is only expected to increase in the future due to an aging population, rising costs of Long-Term Acute Care Hospitals (LTACHs), and wider availability of advanced life-sustaining therapies like hemodialysis, mechanical ventilation, ventricular assist devices, and extracorporeal liver support devices. 

Medical and surgical patients who develop sepsis-related CCI have distinct risk factors and comorbidity profiles, and their clinical trajectory would, therefore, be expected to differ [9,10]. Patients with surgical sepsis typically undergo surgical procedures that lead to inflammation, acute blood loss, prophylactic antibiotic exposure, and secondary surgical complications (surgical site infections, deep venous thromboses with pulmonary thromboembolism, among others) [10,11]. Conversely, critically ill, non-surgical (medical) sepsis patients may have a greater burden of chronic medical comorbidities that affect their post-septic prognosis commensurately.

To gain a better understanding of the clinical trajectory of this patient population with high morbidity and healthcare burden, we conducted an analysis of long-term outcomes in patients with medical CCI following sepsis. We then compared these outcomes with those of patients who developed surgical CCI following sepsis. Given that mechanical ventilation and sepsis are the top two known causes of CCI [5], we focused on this patient population. Due to the rarity of CCI, even in quaternary care medical centers in the United States, we use a case series to describe each cohort of patients. This manuscript is written following guidelines provided by the Case Series Critical Appraisal Tool of the Joanna Briggs Institute (JBI) [12,13].

## 2. Methods

### 2.1. Case Series Description

All septic, critically ill adult patients described in this case series were treated in the medical intensive care unit (MICU) or surgical intensive care unit (SICU) at a quaternary care academic medical center between May 2013 and August 2022. Surgical ICU patients were defined as patients who have undergone a prior surgical procedure and were receiving critical care therapy in the SICU at the time of enrollment, whereas medical ICU patients did not undergo a prior surgical procedure during their hospitalization and were receiving critical care therapy in the MICU. For the purposes of this case series, the following inclusion criteria were used: (1) age ≥ 18 years, (2) acute respiratory failure requiring mechanical intubation or bilevel positive airway pressure (BiPAP) at the time of recruitment, and (3) sepsis or septic shock by Sepsis-3 criteria at the time of recruitment [14].

Chronic critical illness (CCI) was defined as an ICU length of stay ≥14 days with evidence of persistent organ dysfunction, determined by using the components of the Sequential Organ Failure Assessment (SOFA) score (cardiovascular SOFA ≥ 1, or score in any other organ system ≥ 2) [15]. Early death was defined as death within 14 days of sepsis diagnosis. Rapid recovery (RAP) was defined as patients who did not meet the criteria for either CCI or early death. The primary outcome was one-year mortality, as determined by chart review and telephone interview using a standardized questionnaire, performed at one year following study enrollment. If the patient was not available or able to participate in the phone interview, the latter was instead conducted via the patient’s healthcare proxy. Secondary outcomes included ECOG/Zubrod score at one year [16] and place of residence (home versus non-home) at one year. Additional secondary outcomes included in-hospital mortality, ICU-free days, ventilator-free days, 30-day ECOG/Zubrod score, and post-ICU disposition. 

### 2.2. Statistical Analysis

Descriptive statistics were used to characterize the study population, with continuous variables reported as mean ± standard deviation and categorical variables reported as counts with proportions. Statistical tests were two-tailed with a level of significance set at α = 0.05. Inter-cohort differences were analyzed using Analysis of Variance or Student’s t-test for continuous variables and Chi-Square Test or Fisher’s exact test for categorical variables. Nonparametric, pair-wise comparisons were performed using Wilcoxon method. The survival curves were calculated using the Kaplan-Meier method with a 95% confidence interval and end point of 365 days after ICU admission. Analyses were performed using Stata 14.5 (StataCorp, College Station, TX, USA). 

## 3. Results

Eighty-two patients met the criteria for medical sepsis, of which 31 (38%) developed CCI (Figure 1A). Seventeen patients met the criteria for surgical sepsis, of which 7 (41%) developed CCI (Figure 1B). Baseline characteristics for patients developing surgical CCI and medical CCI are shown in Table 1 and Table 2, respectively. Organ dysfunction scores during acute illness differed significantly between medical CCI patients and surgical CCI patients (*p* = 0.0017).

We report both short-term (Table 3) and long-term (Table 4) clinical outcomes in septic patients developing CCI. Interestingly, while surgical patients with CCI experienced significantly more ventilator-free days and significantly less residual organ dysfunction at 14 days post-sepsis, one-year outcomes between surgical and medical patients did not differ. One-year survival analyses, stratified by clinical trajectory, are shown for patients with surgical and medical CCI (Figure 2). A direct comparison between one-year survival in surgical versus medical CCI patients is shown in Figure 3. *p*-value was not calculated due to sample size.

Within the confines of our sample size, we performed multivariable regression analyses to correct for clinically relevant variables that may affect long-term survival, including age, APACHE II, Charlson comorbidity index, and presence of septic shock (Table 5).

## 4. Discussion

Patients suffering from CCI experience prolonged disability and impose a disproportionately large burden on healthcare resources. While patients who eventually develop CCI represent only a minority among septic patients, the number of these patients is increasing [2,9]. To reduce the onset and progression of CCI, early and aggressive interventions are paramount. In surgical environments, optimizing preoperative conditions, ensuring meticulous surgical technique, timely source control, and minimizing complications play vital roles [18]. For both surgical and medical patients, early identification and treatment of sepsis, using goal-directed therapies and adhering to sepsis bundles, can curtail the progression of organ dysfunction [19]. Tailored antibiotic therapies, early mobilization, nutritional support, and proactive weaning from mechanical ventilation may further decrease ICU length of stay and CCI risk. Regular multi-disciplinary team discussions can steer individualized care and promote timely interventions. Understanding these patients’ longer-term clinical trajectories becomes imperative.

The primary objective of the present analysis was to describe the trajectory of the most gravely ill patients with sepsis-induced CCI and to examine whether clinical trajectories differed between sepsis of medical versus surgical origin. Given the infrequency of CCI, a descriptive case series was the most feasible way to accomplish this. Our investigation is the first, to our knowledge, to compare medical versus surgical sepsis-induced CCI in this manner. It demonstrated that short-term clinical outcomes were better in patients having surgical CCI, while one-year mortality and the proportion of patients living at home were similar in both groups.

Septic surgical patients who develop CCI often exhibit a well-recognized phenotype known as persistent inflammation, immunosuppression, and catabolism syndrome (PICS) [20,21,22]. Clinical manifestations of PICS include persistent kidney injury, muscle wasting, cognitive dysfunction, and, ultimately, death [20,21,23]. These patients rarely regain their pre-illness functional status [22]. A recent study reported that patients with surgical sepsis requiring critical care therapy experience a 30-day mortality rate of 9.6% (with only 4% of patients dying within 14 days) but that they often experience unfavorable long-term outcomes, including a one-year mortality rate of 20.9% [9]. Of patients who progress to CCI, one-year mortality increases to 41.4% [9]. While the size of our cohort of surgical CCI patients was considerably smaller than that described by Brakenridge et al., our one-year mortality of 42.9% was remarkably similar. Our reported one-year Zubrod score, however, was significantly higher than that previously reported (3.0 ± 2.2 versus 1.4 ± 0.8, respectively) [9]. The proportion of patients living at home at one-year post-enrollment was not reported by the prior study [9]. 

The term CCI is uncommonly encountered in the medical (non-surgical) literature, especially as it relates to sepsis. Studies that focus on medical patients do not have a universally accepted definition of CCI, most likely due to the heterogeneity of the patient population [24]. However, most studies define CCI as an intensive care unit length of stay lasting ≥ 14–21 days, with persistent organ dysfunction (i.e., SOFA score ≥ 2) [24]. Primary causes of CCI include sepsis (63.7% of cases) and mechanical ventilation (72.0% of cases), which is particularly relevant with respect to the recent COVID-19 pandemic [5,24]. Specifically, one investigation that focused on patients having COVID-19 complicated by CCI reported a 28% ICU mortality rate in this patient population [24], which was nonetheless lower than that found in our equivalent cohort (48.4%). This difference may be attributed to a higher severity of illness associated with the COVID-19 patients in the MICU, resulting in higher rates of early death and, therefore, fewer 14-day survivors. Our rate of medical patients on mechanical ventilation and with CCI closely aligns with previously reported rates of up to 50% [5]. 

Being a case series description, our report has several limitations. First, cohort sizes are relatively small, which may limit the generalizability of the findings to a broader population. Generalizability to other critically ill patients may also be reduced by our description of the ‘sickest of the sick’ patients (i.e., septic patients with acute respiratory failure). While the number of patients in this cohort precluded our ability to perform robust statistical queries such as regression analyses, the disproportionate healthcare burden imposed by this small number of severely ill patients nonetheless mandates a better understanding of their clinical trajectory. Secondly, as a retrospective case series, there is a potential for selection bias, as cases were identified based on available medical records. Additionally, the absence of a control group (i.e., critically ill and non-septic patients) restricts our ability to establish causal relationships. Lastly, the study design inherently carries the risk of incomplete or missing data, and although efforts were made to minimize this through meticulous data collection, there is a possibility of data incompleteness or measurement error. These limitations should be considered when interpreting the results of this case series.

## 5. Conclusions

In conclusion, septic patients with acute respiratory failure who develop CCI represent a minority of septic patients but a tremendous healthcare burden. While the infrequency of this condition makes randomized and controlled trials unfeasible, our study adds value to the body of literature about this condition and may assist patients and healthcare providers in better understanding the epidemiology and short- and long-term outcomes of this disease. Long-term outcomes in CCI patients following sepsis of medical versus surgical origin may not differ significantly despite the different clinical profiles of these two patient populations. Rather, the persistence of organ dysfunction for 14 days or more following sepsis onset appears to be the most significant prognostic predictor.

## Figures and Tables

**Figure 1 medicina-59-01617-f001:**
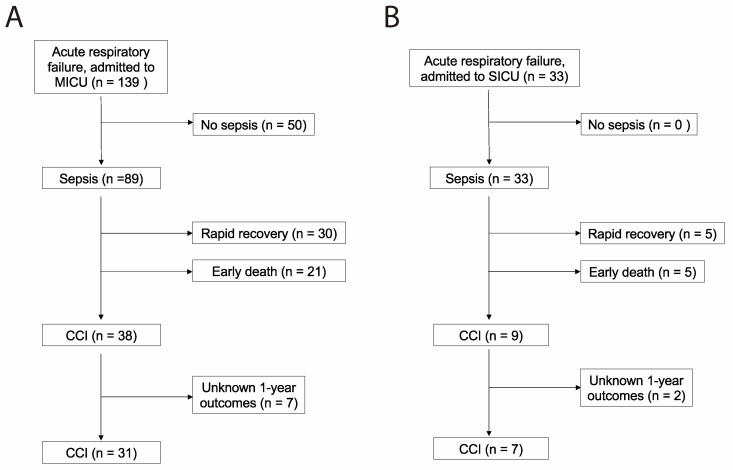
STROBE flow charts describing (**A**) medical versus (**B**) surgical patients with chronic critical illness. STROBE, Strengthening the Reporting of Observational Studies in Epidemiology [17]. MICU = medical intensive care unit; SICU = surgical intensive care unit.

**Figure 2 medicina-59-01617-f002:**
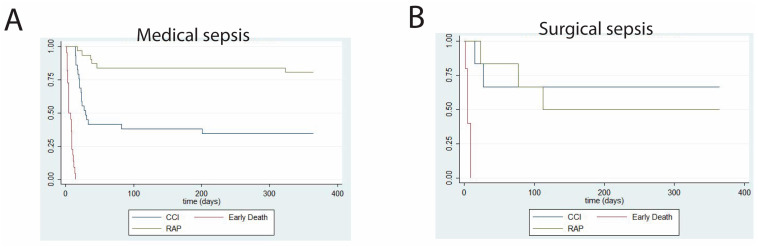
Kaplan–Meier curves showing one-year survival probabilities for patients with (**A**) medical versus (**B**) surgical sepsis and acute respiratory failure. *n =* 83 (medical sepsis) and *n =* 17 (surgical sepsis). CCI = chronic critical illness; early death = death within 14 days of sepsis onset; RAP = rapid recovery.

**Figure 3 medicina-59-01617-f003:**
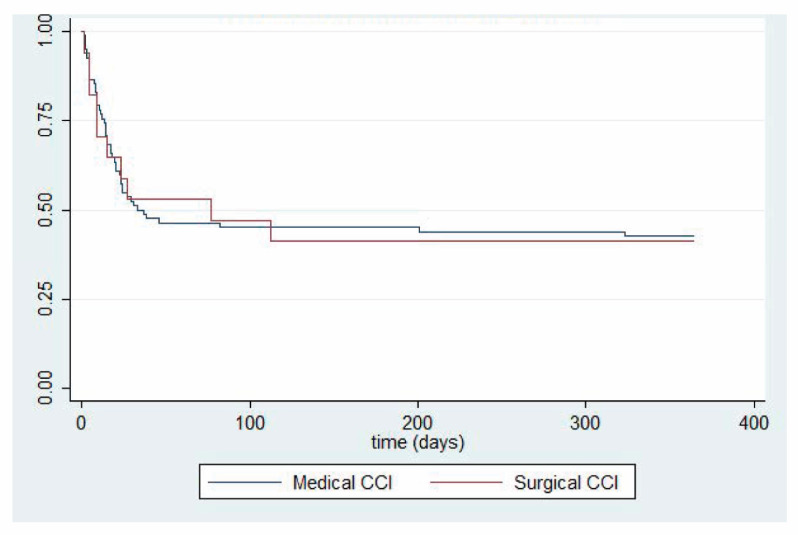
Kaplan–Meier comparing one-year survival probability between patients with medical versus surgical sepsis with acute respiratory failure complicated by chronic critical illness. *n =* 21 (medical sepsis) and *n =* 7 (surgical sepsis). CCI = chronic critical illness.

**Table 1 medicina-59-01617-t001:** Baseline characteristics, by clinical trajectory, for surgical sepsis patients with acute respiratory failure.

Characteristics	RAP (*n* = 5)	CCI (*n* = 7)	Early Death (*n* = 5)	*p* *	*p* **
Male, *n* (%)	3 (60)	5 (71)	2 (40)	0.55	0.68
Age in years, mean ± SD	74.4 ± 16.6	69.7 ± 5.8	56.6 ± 22.3	0.18	0.66
Age ≥ 65 years, *n* (%)	5 (100.0)	6 (85.7)	3 (60.0)	0.24	0.29
BMI, median (25th, 75th percentiles)	25.0 (23.0, 43.2)	35.3 (27.5, 42.7)	33.0 (27.0, 37.4)	0.88	0.68
Charlson comorbidity index score, mean ± SD	5.2 ± 1.5	5.43 ± 2.67	5.8 ± 3.93	0.96	0.78
APACHE II score, mean ± SD	26 ± 9.4	21.9 ± 7.2	24.2 ± 4.3	0.63	0.95
SOFA score, mean ± SD	9 ± 2.8	8.7 ± 3.6	11.6 ± 4.2	0.42	0.95
Septic shock, *n* (%)	2 (40)	2 (28.6)	4 (80)	0.20	0.68
Sepsis source:					
- abdominal, *n* (%)	1 (20)	2 (28.6%)	0 (0)	0.43	0.74
- lung, *n* (%)	1 (20)	4 (57.1)	2 (40)	0.43	0.20
- other, *n* (%)	3 (60)	1 (14.3)	3 (60)	0.17	0.10

* *p*-value comparing the three groups via ANOVA. ** *p*-value comparing RAP and CCI only via two-tailed *t*-test or Chi-squared test. RAP rapid recovery; CCI = chronic critical illness.

**Table 2 medicina-59-01617-t002:** Baseline characteristics, by clinical trajectory, for medical sepsis patients with acute respiratory failure.

Characteristics	RAP (*n* = 30)	CCI (*n* = 31)	Early Death (*n* = 21)	*p* *	*p* **
Male, *n* (%)	19 (63)	17 (55)	12 (57)	0.79	0.50
Age in years, mean ± SD	60.5 ± 16.9	60.4 ± 16.3	62.5 ± 17.6	0.89	0.99
Age ≥ 65 years, *n* (%)	17 (56.7)	12 (38.7)	13 (61.9)	0.20	0.16
BMI, median (25th, 75th percentiles)	30.2 (22.6, 35.5)	32.9 (27.1, 39.8)	28.4 (22.5, 39.0)	0.48	0.19
Charlson comorbidity index score, mean ± SD	3.6 ± 2.4	4.6 ± 3.2	5.3 ± 3.3	0.08	0.06
APACHE II score, mean ± SD	25.9 ± 7.0	28.2 ± 8.9	31.4 ± 7.9	0.07	0.29
SOFA score, mean ± SD	8.9 ± 3.8	12.0 ± 3.6	11.8 ± 3.4	0.0023	0.0017
Septic shock, *n* (%)	7 (23.3)	10 (32.3)	10 (47.6)	0.19	0.44
Sepsis source:					
- abdominal, *n* (%)	2 (6.5)	8 (25.8)	4 (19.1)	0.13	0.04
- lung, *n* (%)	20 (66.7)	18 (58.1)	13 (61.9)	0.79	0.49
- other, *n* (%)	8 (26.7)	5 (16.1)	4 (19.1)	0.58	0.05

* *p*-value comparing the three groups via ANOVA. ** *p*-value comparing RAP and CCI only via two-tailed *t*-test or Chi-squared test. RAP rapid recovery; CCI = chronic critical illness.

**Table 3 medicina-59-01617-t003:** Short-term clinical outcomes in patients developing chronic critical illness.

Short-Term Outcomes	Surgical CCI (*n* = 7)	Medical CCI (*n* = 21)	*p*-Value
28-day mortality, *n* (%)	3 (42.9)	16 (51.6)	0.68
ICU-free days within first 28 days, median (25th, 75th percentiles)	5.0 (3.5, 9.5)	0 (0, 6.0)	0.04
ICU mortality, *n* (%)	3 (42.9)	15 (48.4)	0.79
Hospital LOS, median (25th, 75th percentiles)	23 (16, 28)	24 (20, 35)	0.30
Ventilator-free days *, median (25th, 75th percentiles)	21 (11,24)	3 (0, 12.5)	0.0039
28-day readmission, *n* (%)	2 (28.6)	4 (12.9)	0.30
SOFA score at day 14, mean ± SD	4.3 ± 2.7	8.0 ± 3.9	<0.0001
AKI, *n* (%)	4 (57.1)	18 (58.1)	0.96
Discharge disposition, *n* (%):			
- Discharged home	1 (14.2)	3 (9.7)	0.72
- Long-term acute care facility	2 (28.6)	5 (16.1)	0.44
- Skilled nursing facility	0 (0)	2 (6.5)	0.49
- Hospice care	1 (14.2)	2 (6.5)	0.49

* Ventilator-free days refer to the number of days without mechanical ventilation within 28 days of study enrollment. *p*-values represent analysis via two-tailed *t*-test or Chi-squared test.

**Table 4 medicina-59-01617-t004:** One-year clinical outcomes in patients developing chronic critical illness.

One-Year Outcomes	Surgical CCI (*n* = 7)	Medical CCI (*n* = 21)	*p*-Value
Mortality, *n* (%)	3 (42.9)	21 (67.7)	0.22
Zubrod/ECOG score, mean ± SD	3.0 ± 2.2	4.0 ± 1.7	0.88
Current residence, *n* (%):			
- Home	4 (57.1)	8 (25.8)	0.11
- Long-term acute care facility	0	1 (3.2)	0.63
- Skilled nursing facility	0	0	-
- Hospice care	0	0	-

*p*-values represent analysis via two-tailed *t*-test or Chi-squared test.

**Table 5 medicina-59-01617-t005:** Regression analysis of mortality between surgical CCI and medical CCI patients.

Mortality	Odds Ratio	Std. Err.	z	P.|z|	95% Conf. Interval	
Surgical vs. Medical	4.20	5.00	1.21	0.228	0.41	43.36
Age	0.97	0.03	−0.85	0.395	0.91	1.04
APACHE II	0.91	0.06	−1.39	0.164	0.80	1.04
Charlson comorbidity index	0.82	0.14	−1.13	0.259	0.58	1.16
Septic shock	1.74	1.93	0.50	0.619	0.20	15.36

## Data Availability

The data and materials used in this study are available upon reasonable request from the corresponding author. Restrictions may apply to the availability of certain data sets due to privacy or ethical restrictions.

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
