# Peer review of "Comparing Long-Term Prognosis in Chronic Critically Ill Patients: A Case Series Study of Medical versus Surgical Sepsis"

_medicina, 2023, doi:10.3390/medicina59091617_

Round 1
Reviewer 1 Report
the manuscript is very interesting and deals with an important problem. I would deepen the aspects regarding the prevention of development between the surgical and medical environments. Also do nosocomial infections have a worse prognosis? finally, I recommend quoting this article DOI: 10.1177/2058738419855226.
Author Response
-The manuscript is very interesting and deals with an important problem. I would deepen the aspects regarding the prevention of development between the surgical and medical environments.
Reply: Thank you for this comment. It seems the reviewer is suggesting that we provide more detailed information or discussion about how the onset or progression of CCI might be prevented in both surgical (SICU) and medical (MICU) critical care units. We have added some further information and references in the first part of the ‘Discussion’ section, to address this reviewer’s comment.
-Also do nosocomial infections have a worse prognosis?
Reply: This is a good question. Medical and surgical populations have unique risk factors for nosocomial infection. For example, patients receiving care in a surgical intensive care unit often have surgical wounds that increases the risk of nosocomial infection. Our results suggest that long-term prognosis between medical and surgical patients having CCI are not significantly different, although we did not specifically assess the incidence of nosocomial infection in each population. Therefore, we cannot make any conclusions about the association between incidence of nosocomial infections in each population and the unchanged long-term prognosis between populations.
-Finally, I recommend quoting this article DOI: 10.1177/2058738419855226.
Reply: Thank you, this has been added as reference number 4. We have re-numbered the other references accordingly.
Reviewer 2 Report
In this study, authors investigated a case series study of medical versus surgical sepsis. They describe a cohort of patients having acute respiratory failure and sepsis and requiring critical care therapy in the medical (MICU) or surgical (SICU) critical care units for 14 days. Given the relative infrequency of CCI, they use a case series design to examine mortality, functional status and place of residence (home versus non-home) at one year following their index hospitalization. While surgical patients that develop sepsis-related CCI experience more favorable 30-day outcomes as compared with medical patients, long-term outcomes do not differ significantly between groups. This suggests that reversing established organ dysfunction and functional disability, regardless of etiology, is more challenging compared to preventing these complications at an earlier stage. In general, this study is interesting. Here is one minor comment from this reviewer.
1. Please indicate the statistical analysis method for each table.
Author Response
In this study, authors investigated a case series study of medical versus surgical sepsis. They describe a cohort of patients having acute respiratory failure and sepsis and requiring critical care therapy in the medical (MICU) or surgical (SICU) critical care units for 14 days. Given the relative infrequency of CCI, they use a case series design to examine mortality, functional status and place of residence (home versus non-home) at one year following their index hospitalization. While surgical patients that develop sepsis-related CCI experience more favorable 30-day outcomes as compared with medical patients, long-term outcomes do not differ significantly between groups. This suggests that reversing established organ dysfunction and functional disability, regardless of etiology, is more challenging compared to preventing these complications at an earlier stage. In general, this study is interesting. Here is one minor comment from this reviewer.
Please indicate the statistical analysis method for each table.
Reply: Thank you. Designations for analysis via ANOVA versus t-test have been added.
Reviewer 3 Report
In a case series “Comparing Long-Term Prognosis in Chronic Critically Ill Patients: A Case Series Study of Medical versus Surgical Sepsis” the authors investigated whether source of sepsis (medical versus surgical) affected clinical trajectory and prognosis in the patients. In their cohort they included n=31 medical patients developing chronic critical illness (CCI) and n=7 of surgical patients. They have concluded that surgical patients that develop sepsis-related CCI experience have more favorable 30-day outcomes as compared with medical patients, while long-term outcomes do not differ significantly between groups. The authors suggested that reversing established organ dysfunction and functional disability, regardless of etiology, is more challenging compared to preventing these complications at an earlier stage.
The study is interesting, and the topic is very relevant, but number of patients is small, making conclusions very limited and biased. The strength of the study is comparison between the medical and surgical patients in short and long term (one year). The study has multiple limitations that are acknowledged by the authors.
Comments:
1. The study examined small group of the patients admitted in medical or surgical ICU. The number of the patients in surgical group is particularly small, making the comparison between the groups challenging.
2. Figure 1. – please increase size of the letters.
3. English should be slightly improved.
English should be slightly improved.
Author Response
In a case series “Comparing Long-Term Prognosis in Chronic Critically Ill Patients: A Case Series Study of Medical versus Surgical Sepsis” the authors investigated whether source of sepsis (medical versus surgical) affected clinical trajectory and prognosis in the patients. In their cohort they included n=31 medical patients developing chronic critical illness (CCI) and n=7 of surgical patients. They have concluded that surgical patients that develop sepsis-related CCI experience have more favorable 30-day outcomes as compared with medical patients, while long-term outcomes do not differ significantly between groups. The authors suggested that reversing established organ dysfunction and functional disability, regardless of etiology, is more challenging compared to preventing these complications at an earlier stage.
The study is interesting, and the topic is very relevant, but number of patients is small, making conclusions very limited and biased. The strength of the study is comparison between the medical and surgical patients in short and long term (one year). The study has multiple limitations that are acknowledged by the authors.
Comments:
- The study examined small group of the patients admitted in medical or surgical ICU. The number of the patients in surgical group is particularly small, making the comparison between the groups challenging.
Reply: This is a valid point. Our choice of a 'case series' format and the restrained statistical analysis stem primarily from this reason. While a more sizable patient cohort would undeniably bolster the study's significance and permit a more comprehensive statistical evaluation, amassing data from such a cohort would also demand several years.
- Figure 1. – please increase size of the letters.
Reply: Thank you, this change has been made.
- English should be slightly improved.
Reply: Thank you for the comment, although all authors are native English speakers with extensive publication histories for which language editing services have never been required in the past. We would be happy to fix any minor wording or to provide further clarity if you could point us to specific lines in the manuscript.